# Epigenetic Modifications in Prostate Cancer Metastasis and Microenvironment

**DOI:** 10.3390/cancers15082243

**Published:** 2023-04-11

**Authors:** Shouyi Zhang, Tao Shen, Yu Zeng

**Affiliations:** 1Department of Urology, the Cancer Hospital of Dalian University of Technology & Liaoning Cancer Hospital, Shenyang 110042, China; 2Department of Urology, Second Affiliated Hospital of Shenyang Medical College, No. 20 Beijiu Road, Heping District, Shenyang 110001, China

**Keywords:** epigenetic modifications, prostate cancer, tumor microenvironment, tumor remodeling

## Abstract

**Simple Summary:**

Epigenetics, which leads to specific differentiation events and determines gene expression states, has been recognized as a developmental landscape. Epigenetic modifications are widespread in mammals. Dysregulation of epigenetic modifications is closely related to the occurrence of diseases, especially cancer. The relationship between epigenetic modification and cancer has been widely studied. This review aims to summarize the different epigenetic modifications events that occur in prostate cancer and improve our understanding of the biological role of epigenetic modifications in prostate cancer metastasis.

**Abstract:**

The gradual evolution of prostate tissue from benign tumor to malignant lesion or distant metastasis is driven by intracellular epigenetic changes and the tumor microenvironment remodeling. With the continuous study of epigenetic modifications, these tumor-driving forces are being discovered and are providing new treatments for cancer. Here we introduce the classification of epigenetic modification and highlight the role of epigenetic modification in tumor remodeling and communication of the tumor microenvironment.

## 1. Introduction

The concept of epigenetics was developed by Conrad Waddington and was originally used to describe heritable changes in cell phenotypes independent of DNA sequence [1]. At present, a variety of epigenetic modifications have been identified, including at least seventeen kinds of DNA modifications and 160 kinds of RNA modifications [2,3]. As a rapid and dynamic method to regulate cell behavior without altering DNA sequence, epigenetic modification plays an important role in cell remodeling during communication between cells and the environment where they exist [4,5]. The most important DNA modification is DNA methylation (5mC), which mainly affects DNA–protein interactions in the double helix structure but does not affect Watson–Crick pairing [6]. It has been shown that global alterations to the DNA methylation landscape contribute to alterations in the transcriptome and deregulation of cellular pathways [7]. At the same time, DNA methylation provides a driving force for tumor cell remodeling. For example, in early-stage prostate cancer, genes such as APC, RASSF1, GSTP1 undergo DNA hypermethylation [8]. In particular, GSTP1 shows promoter hypermethylation in approximately 90% of PCa and 70% of prostatic intraepithelial neoplasia (PIN) patients [9]. In addition to DNA methylation, chromosome-related regulatory means such as histone modification [10], nucleosome localization [11], and higher-order chromatin organization [12] play important roles in the phenotypic transformation of prostate cancer. Recently, due to the breakthrough of high-throughput sequencing technology, the research for detecting RNA modifications has exploded [13]. The main mRNA modifications include N6-methyladenosine (m^6^A), N6, 2′-o-dimethyladenosine (m^6^Am), N1-methyladenosine (m^1^A), 5-methylcytosine (m^5^C), N4-acetylcytidine (ac^4^C), pseuduridine (Ψ), N7-methylguanosine (m^7^G) and so on. m^6^A methylation is the most abundant mRNA modification in mammals and is also crucial in prostate cancer [14]. As a result of copy number variations (CNVs) events in PCa, m6A methylation is altered at high levels in most prostate cancer patients and is considerably associated with a recurrence-free survival of prostate cancer [15]. In addition, m6A-related regulatory molecules also experience different events in prostate cancer, and most are associated with poor prognosis [14]. The dysregulation of RNA modification is based on the regulation of DNA methylation on the one hand [16], and is closely related to intercellular communication on the other hand [7]. Additionally, non-coding RNAs (ncRNAs) raised from RNA modification can also participate in the communication between tumor cells and microenvironment [17]. Here, we discuss the role of different epigenetic modifications in the development of prostate cancer (PCa), especially when they metastasize, as well as the application prospect of epigenetic regulation in cancer therapy.

## 2. Chromatin-Related Regulation

### 2.1. DNA Methylation

DNA methylation is a process associated with chromatin closure and repression of gene expression, and is the addition of a methyl group (CH_3_) at the 5-carbon position of cytosine (5-methylcytosine, 5mC) [7]. DNA methylation often occurs on CpG islands, a region of more than 200 bases formed by aggregation of CpG dinucleotides [18], and the methylation process is mainly mediated by DNMT family enzymes. The enzyme catalyzes the transfer of methyl groups from S-adenosylmethionine to DNA, and it is currently thought that only DNMT1 (DNA methylases 1), DNMT3A, and DNMT3B contain methyltransferase activity. Methylation of 5′ carbocytidine (5mC) in CpG dinucleotides in gene promoters is considered to be the most direct epigenetic mechanism to maintain gene silencing [19]. Therefore, the dysregulation of DNA methylation under the influence of environmental and genetic factors has long been considered as a major driver of cancer development. 5mC can be oxidized to 5-hydroxymethylcytosine (5hmC) by the ten-eleven translocated (TET) family enzymes, resulting in a reversal of methylation. Unlike 5mC modification, the number of 5hmC modifications represent only a small fraction of the total DNA methylation modifications, and are enriched in the transcription active region. In fact, most cancer-related sites undergo epigenetic changes early in prostate cancer development [20]. DNA methylation profiles between adjacent benign and noncancerous prostate tissues are also significantly different [21,22]. Thus, gene hypermethylation caused by improper catalysis of DNMTs may represent an early onset event for prostate cancer development [23]. It has been found that 5hmC modification was enriched in the binding sites of AR, FOXA1 and HOXB13, which are the major driver genes for development of PCa, especially for metastatic castration resistant PCa (mCRPC) [24]. These hint that 5hmC may be involved in gene reprogramming during the development of metastatic PCa.

### 2.2. Histone Modification

Core histones in cells include H2A, H2B, H3, and H4. Histones are predominantly globular, and their N-terminal tails are the site of post-transcriptional modifications, including acetylation, methylation, phosphorylation, ubiquitination, SUMO, and ADP-ribosylation [25,26]. Histone modifications play an important role in transcriptional regulation, DNA replication, DNA repair, alternative splicing, and chromosomal agglutination [27,28]. Histone modification-related enzymes thought to be associated with prostate cancer progression include histone methyltransferase EZH2 [29], lysine-specific demethylase 1 (LSD1) [30], histone methyltransferase SET9 [31], and others. Different histone modification patterns also exist in primary prostate cancer, including acetylated histone H3 lysine 9, acetylated histone H3 lysine 18 (H3K18Ac), H4K12Ac, di-methylated H4 arginine 3 (H4R3me2) and di-methylated H3 lysine 4 (H3K4me2) [32]. High levels of H3K18Ac and H3K4me2 are associated with a risk of prostate cancer recurrence [33].

### 2.3. Nucleosome Positioning

Nucleosomes regulate gene expression by participating in polymerases that inhibit transcript elongation and prevent transcription factors from entering the transcriptional start site (TSS). The precise localization of nucleosomes is affected by different histone variants [34]. Nucleosome remodeling is regulated by DNA methylation [35], histone modification [34], chromatin remodeling complexes [18], etc. Nucleosomes are remodeled in a manner dependent on ATP hydrolysis by chromatin remodeling complexes, which can be divided into four families (SWI/SNF, ISWI, CHD, and INO80) [18]. CDH1 deletion in prostate cancer is strongly associated with early recurrence of prostate cancer, a high Gleason score, and advanced tumor stage [36,37]. In addition, the localization of nucleosomes will be changed after androgen therapy. Based on this model of androgen-responsive nucleosome behavior, researchers can accurately predict the binding of disease-associated transcription factors during treatment [11].

### 2.4. Higher-Order Chromatin Organization

Unlike local chromatin regulation, the ‘higher order’ level of chromatin regulation occurs at a more global level, involving changes in nuclear localization, associations or larger chromatin regions with repressive compartments, such as the nuclear periphery or pericentromeric heterochromatin, and large-scale changes in DNA structure, such as the formation of DNA loops or locus contraction [38]. There is controversy over the status of the higher-order chromatin organizations in prostate cancer. It has been suggested that the higher-order chromatin organization status in prostate cancer is stable relative to benign tissues, and the changes in local chromatin interactions mediate the development of prostate cancer [12]. Conversely, other studies suggest that the higher-order chromatin organization is disordered and co-exists with the change of epigenetic in prostate cancer [39], among which the factor CAF-1, which is involved in the conformation of the higher-order chromatin organization, is thought to be significantly associated with adverse biological behavior in prostate cancer [40]. In addition, PCa-related risk SNPs containing CREs (rCREs) screened by CRISPRi were shown to have interactions with CTCF. As a key regulator of the three-dimensional (3D) genome architecture, CTCF-mediated 3D chromatin interactions may lead to dysregulation of neighboring genes in Pca. The interaction between DNA-methylation-dependent CTCF deposition and rCREs indicates that the DNA methylation that affects the 3D architecture contributes to causes of prostate cancer [41].

## 3. RNA Modification

### 3.1. m^6^A

As a widespread RNA modification in mammals, the abundance of m^6^A modification is the highest among all kinds of RNA modifications [42]. m^6^A modification has regulatory effects on a variety of RNA processes, such as RNA alternative splicing, translation, translocation, and stability [43,44]. The modification site of m^6^A is generally located in the region consisting of a conserved motif DRACH (D = A/G/U, R = A/G, H = A/U/C). The process of m^6^A includes recruiting methyltransferase (METTL3/14) to ‘write’ methylation, attracting m^6^A binding protein (YTHDF1/2/3, YTHDC1/2, HNRNPA2B1, IGF2BP1/2/3) to ‘read’ methylation, and finally the methylation is ‘erased’ by demethylase (ALKBH5, FTO) [45]. Therefore, m^6^A modification is a dynamic process. In prostate cancer tissue, an increasing level of m^6^A methylation has been shown, probably due to the CNVs (DNA copy number loss) events of demethylase genes, as well as the overexpression of methyltransferases [15]. At the same time, m^6^A modification has also been found to be associated with therapy resistance of immunotherapy or androgen receptor inhibitor (ARPI) in PCa [46,47].

### 3.2. m^6^Am

m^6^Am shares the same demethylase (FTO) with m^6^A modification [48]. It has been shown that if the first nucleotide adjacent to the m^7^G cap is 2′-O-methyladenosine (Am), it can be further methylated at position N6 to form m^6^Am [49]. To date, phosphorylated CTD interactor 1 (PCIF1) has been identified as the only known m^6^Am-programmed protein [50]. However, the research on cancer-related m^6^Am is still limited, and it requires a breakthrough at the technical level to distinguish m^6^A and m^6^Am by using conventional experimental means [51]. So far, it has been shown that the expression level of PCIF1 in prostate cancer tissue is significantly different compared with those adjacent normal tissues or infilled immune cells [52].

### 3.3. m^1^A

There are similar ‘readers’ and ‘erasers’ between m^1^A and m^6^A. The ‘writers’ of m^1^A are TRMT6/61A/61B/10C and NML [53]. It has been shown that the total level of m^1^A modification is about one-tenth of m^6^A modification [54,55]. The modification on the majority of m^1^A sites is at the ultra-low level, except for a single site in the mitochondrial encoding gene ND5, in which it shows high m^1^A methylation levels [42]. Although the role of m^1^A in prostate cancer has not been reported yet, some studies have shown that nitrogen atoms can undergo Dimroth rearrangement from m^1^A to m^6^A in the alkaline environment [56], suggesting a dynamic regulation between m^1^A and m^6^A under certain conditions.

### 3.4. m^5^C

m^5^C is conserved and widespread, especially in rRNA and tRNA [57]. m^5^C is catalyzed by the NOL1/NOP2/Nsun family, DNMT2 and TRDMT1 [58], recognized by YBX1 and ALYREF, and finally ‘erased’ by the TET family (TET1/2/3) and ALKBH1 [59]. It is known that there is high frequency of copy number deletions for many m^5^C regulators in prostate adenocarcinoma, such as ALYREF, NSUN5, DNMT3A, and NSUN2, whereas a high frequency of copy number amplifications are found in some other m^5^C regulators. Seventeen m^5^C regulators have been identified with different expression levels in normal or tumor tissues. m^5^C ‘writers’ and ‘readers’ show high expression, whereas most m^5^C ‘erasers’ show low expression in tumor tissues. m^5^C modification level is significantly associated with older patients, higher Gleason scores, advanced T stages, advanced N stages and higher risk scores. Activation of m^5^C modification may promote tumorigenesis and progression of PRAD (prostate adenocarcinoma) [60]. According to the bioinformatics model, the functions of m^5^C-regulated genes are potentially related to the tumor immune biological process, and they also have a potential role in remolding the tumor microenvironment [61].

### 3.5. ac^4^C

ac^4^C is a cytidine modification widely distributed in non-coding and coding RNA in human cells; it is highly enriched near the translation initiation codon [62]. NAT10 is the only one cytosine acetyltransferase identified at present. The ac^4^C-modified RNA has the characteristics of a longer half-life and a higher translation efficiency than non-ac^4^C-modified RNA [63]. The specific modification mechanism of ac^4^C is not clear, but the expression of NAT10 in prostate cancer is associated with lymph node metastasis and high Gleason scores [64].

### 3.6. Ψ

Ψ is generated by isomerization of the C-C glycoside of the uridine base. The conversion of uridine to Ψ can be formed by RNA-dependent and RNA-independent catalysis, mediated by Box H/ACA small nucleolar RNAs (snoRNAs) and Ψ synthases (PUSs), respectively [65]. The presence of Ψ in mRNA affects the local secondary structure and protein coding potential [62]. It has been shown that Ψ plays a role in affecting mRNA half-life, alternative splicing and RNA stability [62,66]. The most common mutated form of pseuduridine synthase NAP57 is the formation of the NAP57·SHQ1 complex [67]. Interestingly, in integrative genomic profiling studies of prostate cancer, both of the chromosome 3p regions containing SHQ1 and the androgen-driven serine protease-transcription factor fusion TMPRSS 2-ERG have been identified as a tumor inhibitor [68]. Higher Ψ levels are also positively correlated with the progression of prostate cancer, not only in terms of somatic mutations [69,70,71].

### 3.7. m^7^G

N7-methylguanosine (m^7^G) is also a common RNA modification, and the main methyltransferases are METTL1, RNMT, TRMT112, etc. [72]. At present, it is believed that m^7^G has a potential role in regulating the tumor immune microenvironment and metastasis of prostate cancer [73].

Finally, we summarize the epigenetic modification events that occur in prostate cancer (Table 1).

## 4. Epigenetic Modifications in PCa

As we know, the development of prostate cancers is largely driven by AR signaling. In PCa, AR cistrome (the universe of AR-binding sites) undergoes massive reprogramming, leading to the repeated gain and loss of AR-binding sites [74]. In contrast, the reprogramming degrees of the prostate-cancer-specific enhancers FOXA1 and HOXB13 are significantly reduced during the disease progression, and both of them are pre-bound to the AR-binding sites. Therefore, the reprogrammed AR sites are not formed de novo, but are AR-bound to these enhancers. As a result, the degree of DNA methylation at these sites is relatively low, and the mutation burden at these AR-binding sites is also increased [75]. In the case of androgen depletion, AR enhancer modules TMPRSS2, KLK2, and KLK3 show nucleosome-depleted regions (NDRs). In the absence of androgen ligands, AR enhancers exist in an equilibrium, and a proportion of these enhancer modules display NDR status [76]. This “receptive” enhancer activation mode can activate AR for a short time at low ligand concentrations, and also reflects the rapid response of epigenetic regulation to cellular adaptation. It is supposed that the dynamic changes of the AR gene meet the needs of prostate cancer evolution, and the continuous progression of prostate cancer also provides the motivation for AR remodeling. This also reflects the driving role of DNA methylation in the remodeling process of tumor cells.

Circulating tumor cells (CTCs), as the primitive metastasis cells, undergo a remodeling process when entering blood compared with orthotopic tumor cells. This process is defined as EMT. Epigenetic regulation is an important method of regulating EMT [77,78]. For example, in metastatic prostate cancer, the depletion of histone methyltransferase (MMSET) leads to transcription dysregulation, and further regulates EMT and invasion of tumor cells by regulating TWIST1 [79]. Another piece of evidence is that both GSTP1 and RASSF1A show high DNA methylation in circulating tumor cells or paired plasma-derived exosomes of mCRPC [80] and some genes related to androgen synthesis are also hypermethylated (such as CYP11A1, CYP11B1, CYP17A1 and CYP19A1) [81]. As far as we know, GSTP1 has a wide range of physiological functions, and plays an important role in anti-DNA damage, oxidative stress, cell proliferation and death [82]. RASSF1A is a well-known tumor suppressor gene, which crosslinks with many signaling pathways and coordinates multiple cellular processes [83]. The methylation of the promoter region of GSTP1 and RASSF1A represents not only a common feature of tumor cells that escape in situ, but also a plastic change of tumor cells to adapt to the new environment after colonization. These changes are inseparable from the effect of DNA methylation modification. In addition, in advanced prostate cancer, different nucleosome localization models in circulating tumor DNA (ctDNA) also have a significant impact on different phenotypes of prostate cancer. Among them, AR, ASCL1, HOXB13, HNF4G, and GATA2 have been identified in the activity of key phenotype-defining transcriptional regulators from ctDNA [84]. At the same time, research has accurately predicted different clinical phenotypes of prostate cancer by identifying ctDNA and establishing prediction models, which is of great significance for accurate oncology diagnosis. In summary, dynamic epigenetic modifications in prostate cancer provide great help to understand different cancer types and the progression of cancer.

## 5. Epigenetic Modifications in Tumor Microenvironment (TME)

In general, the tumor microenvironment provides tumor cells with inflammatory cytokines, angiogenesis factors, and extracellular matrix (ECM) proteins to create niches for tumor cells [85]. In the metastatic microenvironment, cancer cells acquire stem-like properties through paracrine interactions between cancer-associated fibroblasts (CAF) and cancer cells, which include IL-6 [86]. CHD1 has been shown to active NF-κB in PTEN-deficient prostate cancer cells, which in turn promotes the secretion of IL-6. Mechanistically, CHD1 is stable in PTEN-null prostate cancer cells and interacts with the active epigenetic marker trimethylation of H3K4me3 in the IL-6 gene [87]. GDF15 (MIC-1), a member of the TGFβ/BMP family in tumor stroma, was found to be downregulated in BPH and increased in some PCa samples [88]. Ectopic expression of MIC-1 in fibroblasts leads to a significant paracrine effect on prostate cancer cell migration, invasion and tumor growth [89]. Although the underlying mechanisms have not been reported, a study based on the urine sample analysis of DNA in bladder cancer showed that the methylation level of GDF15 was higher than normal tissue [90]. Thus, there may be a close relationship between MIC-1 alterations and DNA methylation in the TME. In addition, lactic acid secreted by CAF increases the expression of genes involved in lipid metabolism in PCa cells, thereby promoting the growth and transfer of PCa. In this process, CAF enhances PCa intracellular lipid accumulation and provides acetyl moieties for histone acetylation, establishing a regulatory loop between metabolites and epigenetic modification. This work shows that stromal-derived tumor metabolic changes stimulate epigenetic rewiring, and foster metastatic potential in prostate cancer [91].

It has been well known that blood vessels in the tumor microenvironment are very important for tumor growth and distant metastasis. The functional characteristics and cytology between tumor endothelial cells (TEC) and normal endothelial cells (NEC) in the microenvironment are different [92]. The CXCR4/CXCL12 axis has been identified as an anti-angiogenic target that affects the composition of TME and the characteristics of the PCa metastatic vascular network [93]. CXCL12 is a key chemokine in many homeostatic processes such as angiogenesis, inflammation and leukocyte migration, and its active form is strictly regulated and controlled by upstream molecules [94]. In humans, CXCL12 has six kinds of splice variants with specific tissue distribution and properties [95,96]. For example, CXCl12γ has shown weak in vitro chemotaxis and less activation of CXCr4-mediated signaling in all cleaved variants of CXCL12 [97]. This suggests that mRNA splicing regulation based on RNA post-transcription regulation has an important influence on the specific function and activity of CXCL12. As the receptor of CXCL12, CXCR4 is also dysregulated through epigenetic regulation [98]. For example, miR-494-3p has been shown to inhibit CXCR4 expression after transcription processes in prostate cancer [99]; lncRNA UCA1 activates CXCR4 through inhibiting the activity of miR-204 to promote the progression of prostate cancer [100]. In addition, the gene expression of CXCR4 is strongly affected by methylation modification. DNA hypomethylation in the CXCR4 gene has been observed in breast cancer [101], colorectal cancer [102], pancreatic cancer [103] and melanoma [104], and this is associated with tumor progression. Although it has not been reported in prostate cancer, there is evidence that CXCR4 is affected by epigenetic modification in metastatic prostate cancer. In PCa bone metastasis, ac-KLF5 (acetylated KLF5) up-regulates CXCR4 expression through histone acetylation in the promoter region of CXCR4 gene, and further promotes IL-11 secretion, osteoclast differentiation and the regulation of tumor cell plasticity. In addition, the combination of docetaxel and plerixafor (CXCR4 inhibitor) effectively inhibits ac-KLF5-induced bone metastatic lesions and restores the sensitivity to docetaxel in ac-KLF5-expressing tumors [105]. Meanwhile, research also demonstrates that epigenetic modification of the CXCR4 promoter plays a crucial role in the development of bone metastasis in prostate cancer and tumor microenvironment-related chemotaxis.

Recently, ncRNA has shown many important functions in tumor development, and the production of ncRNA also depends on RNA modification [106,107,108]. As one of the most common RNA modifications, m^6^A methylation is widely involved in the cleavage and maturation of ncRNA [109]. At the same time, ncRNAs can also affect the modification process of m^6^A by participating in the binding of m^6^A regulatory molecules to target RNA or regulating m^6^A regulatory molecules [110,111,112]. ncRNAs may play a role in cell communication in promoting prostate cancer metastasis. For example, delta-like 1 homolog-deiodinase, iodothyronine 3 (DLK1-DIO3) clusters were elevated in the serum of metastasis PCa. It has been found that miR-154 and miR-379 promote prostate cancer EMT and bone metastasis by targeting the genes of STAG2 and RSU1 [113]. In addition, ectopic overexpression of miR-409 in normal prostate fibroblasts has also been found to induce tumor-associated stroma-like phenotype and promote EMT of tumor cells [114]. Except for the RNA modifications effect, miRNAs also interfere with histone modifications. EZH2 targets the metastasis suppressor RKIP promoter in prostate cancer and negatively regulates RKIP transcription by inhibiting histone modifications. miR-101 can interfere with this process by down-regulating the expression of EZH2 and then affect the invasion or metastasis of PCa [115]. In this context, ncRNAs not only enrich the means of communication between tumor and stroma, but also expands the interaction between RNA and DNA epigenetic modification [116,117].

The most common metastatic site of PCa is bone, which causes both osteolytic and osteogenic alterations. In the bone microenvironment, there are multiple growth factors including TGFβ, FGF, IGF, and BMP-2. They are not only capable of stimulating the growth of metastatic cancer cells, but also induce the production of bone resorptive factors from tumor cells [118]. It has been found that the promoter methylation levels of APC, TGFb2 and RASSF2001A in prostate cancer are related to Gleason score and pathological stage [119]. During bone metastasis, prostate cancer cells target the hematopoietic stem cell (HSC) niche in bone marrow. Studies have shown that osteoblasts in the HSC niche induce the expression of TBK1 in PCa cells, thereby promoting the maintenance of tumor quiescence, chemoresistance and cancer stem cell characteristics [120]. Although there is no report for this mechanism, it can be found in another study that m^6^A modification mediates immune microenvironment regulation by regulating the ALKBH5/TBK1/IRF3 pathway in neck squamous cell carcinoma [121]. In addition, m^6^A modification has also been reported to be associated with bone or lung metastasis. In this study, the wild type lncRNA NEAT1-1 was more likely to induce bone or lung metastasis, while the m^6^A mutation lncRNA NEAT1-1 was shown to be less likely to form metastases. Mechanically speaking, lncRNA NEAT1-1 recruits CYCLINL1 and CDK9 to the RUNX2 promoter through RNA–DNA interaction, thereby increasing RUNX2 expression. However, the m^6^A mutation lncRNA NEAT1-1 cannot effectively promote RUNX2 expression [122]. ncRNA also plays a role in the process of communication between tumor cells and bone microenvironment. For example, eight kinds of miRNAs have been identified as highly expressed in PCa exosomes and extensively induce osteoblastic lesions. Among them, miR-940 promotes osteogenic differentiation of human mesenchymal stem cells by targeting ARHGAP1 and FAM134A [123]. Interestingly, the osteolytic phenotype breast cancer cell line MDA-MB-231 cells also developed extensive osteoblastic lesions after overexpression of miR-940. Thus, ncRNAs play an important role in the communication between tumor cells and TME. In summary, epigenetic modification plays an important role in the relationship between tumor adaptation and microenvironment remodeling (Figure 1).

Our group has been devoted to the study of RNA modification and metastatic microenvironment. At first, we found that a cold shock protein, RBM3, is upregulated in in situ tumors and downregulated in metastatic prostate cancer. Interestingly, we found that high expression of RBM3 significantly affected the stem-like properties of prostate cancer cells. Through mechanistic studies, we found that RBM3 significantly reduced stem-like properties of tumor cells by inhibiting alternative splicing of CD44v8-10 [124]. This not only leads to alternative splicing, but RBM3 also plays a role in affecting translation efficiency and RNA stability [125,126,127,128], which aroused our interest. In our study, the RNA functional cluster of CLIP-seq suggested that RBM3 might bind to CTNNB1 (beta-catenin mRNA). Wnt/β-catenin plays an important role in bone development and the colonization of bone metastases [129,130,131], and the bone microenvironment also confers tumor cells with stemness and plastic characteristics [132]. Further, we found that RBM3 upregulated m^6^A methylation on CTNNB1 in a METTL3-dependent manner, resulting in decreased stability of CTNNB1 mRNA, thus affecting the adaptive survival of prostate cancer in the bone microenvironment [133]. Thus, our study suggests that the protein function of stress response proteins such as RBM3 have significantly changed after the change of tumor microenvironment, and they are potentially involved in epigenetic modifications by binding to RNA or DNA in response to the changes of the microenvironment. In the above section, we discussed the role of epigenetic modifications in the microenvironment and metastasis of PCa. In general, epigenetic regulation is not only reflected in the adaptive changes of tumor cells, but also has effects on cell metabolism, communication, microenvironment remodeling and other aspects. Unlike normal tissue, in cancer, the dysregulation of epigenetic modifications seems to be a manner for cancer cells to live. Similar to the law of entropy generation, a single tumor cell’s disorder can gradually evolve into a group of cells and even parts of tissues and organs, and eventually dies. The same is true of epigenetic regulation, which is at various stages of the central dogma, and errors at one stage can cause overall loss of control and eventually lead to cancerization. Therefore, it is crucial to understand the processes of epigenetic regulation in cancer. These help us to understand the mechanisms on which cancer depends and to eliminate them.

## 6. Epigenetic Modifications in Cancer Therapy

At present, 99% of oral drugs are targeted to pathogenic proteins. Target drugs have brought patients periodic remission, but also brought new problems of drug resistance. In the human genome, about 1.5% of the DNA sequence encodes proteins, of which disease-related proteins account for only about 15%. The richness of the genome and transcriptome sequences makes up for the shortcomings of the proteome. Various treatments of epigenetic modification have appeared with the increase of epigenetic-modification-related studies. Recently, drugs related to histone modification have been approved, for example four histone deacetylase (HDAC) inhibitors, namely vorinostat, romidepsin, panobinostat and belinostat [134]. In addition, some highly selective methyltransferase inhibitors have achieved remarkable results in vivo. For example, highly selective methyltransferase inhibitors of EZH2, GSK126 and EPZ6438 can restore androgen receptor expression and hormone therapy sensitivity. They also significantly inhibit tumor growth when combined with enzalutamide [135]. Targeting pathogenic proteins has certain limitations, but there are more possibilities for targeting non-coding RNA and methylation/acetylation epigenetic regulation. Currently, researchers have developed platforms for the delivery of specific RNAs [136]. This new RNA-based therapy has brought a major change in the development of drugs. RNA therapy has the natural advantages of low cost, short development cycle and rapid effect, but there are also some problems to be improved.

## 7. Conclusions and Perspectives

The genomic instability of tumor cells is increased with the deterioration of tumors and the pressure of therapeutic intervention. Molecular characteristics in metastatic tumors are often different from primary tumors; this requires us to explore the mechanisms of this change. As C.H. Waddington envisioned, the contours of the cell epigenetic landscape determine the direction of its lineage. Prostate cancer progression and metastasis may reactivate previously developed pathways in specific lineages through epigenetic modifications, which lead to dedifferentiation. Tumor cells often adapt to the change of the environment and stress through DNA methylation, while RNA modification plays a more significant role in the communication between the tumor and the microenvironment. Genetic mutations promote tumor cell adaptive remodeling, which often makes them more malignant and less treatment sensitive. A deeper understanding of the interplay between epigenetic modifications and the tumor metastatic environment will provide more mechanistic insights for therapy.

## Figures and Tables

**Figure 1 cancers-15-02243-f001:**
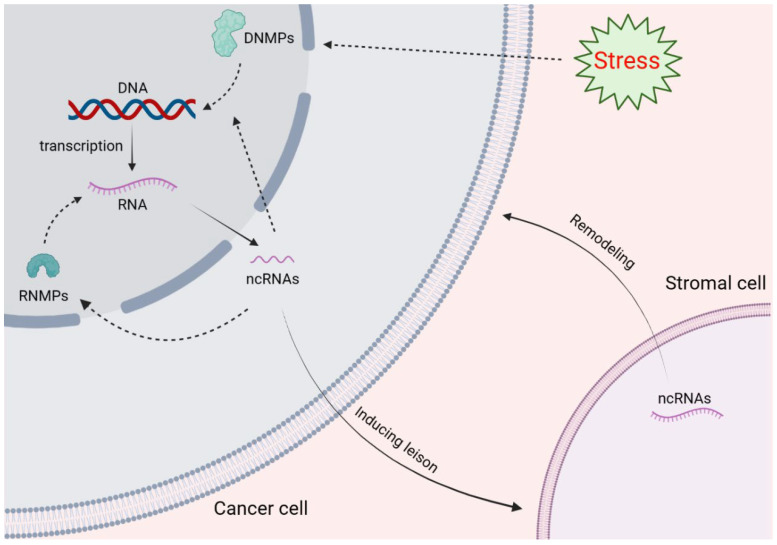
Schematic diagram of epigenetic regulation in the microenvironment. To respond to external stresses, tumor cells are usually regulated in a manner of chromatin regulation, such as DNA methylation, histone modifications, nucleosome localization, and changes in high-grade chromatin conformation. This process not only affects the gene remodeling of tumor cells, but also has a huge effect on cell metabolism and epitranscription. Similarly, as an intermediate process between “response” and “effect”, RNA modification also plays an essential role in tumor cell adaptation and survival. The ncRNAs generated under RNA modification are not only involved in the communication between cancer cells and stromal cells, but are also involved in the process of epigenetic modification. DNMPs—DNA modification proteins; RNMPs—RNA modification proteins.

**Table 1 cancers-15-02243-t001:** Epigenetic modification events in prostate cancer.

Modification	Structure	Molecular	Events on PCa	Experimental Evidence	Reference (PMID)
**m^6^A**	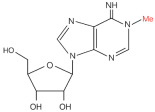	Writer: METTL3/14 Reader: YTHDF1/2/3, YTHDC1/2 Eraser: ALKBH5, FTO	High expression: METTL3, RBM15, HNRNPA2B1 Low expression: ALKBH5, FTO	Transcriptome information and gene-level alteration data from The Cancer Genome Atlas datasets	32710725
**m^6^Am**	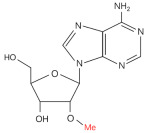	Writer: PCIF1 Reader: — Eraser: FTO	PCIF1 gene deep deletion	TCGA database and GTEx database	34888238
**m^1^A**	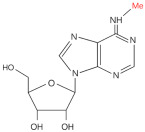	Writer: TRMT6/61A/10C Reader: YTHDF1/2 Eraser: FTO, ALKBH1/3	—	—	—
**m^5^C**	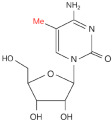	Writer: NSUN2, TRDMT1 Reader: YBX1, ALYREF Eraser: TET1/2/3	High expression: NSUN2/5, DNMT3A/3B, ALYREF, TET1 Low expression: TET2	TCGA and GEO databases	35603206
**ac^4^C**	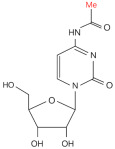	Writer: NAT10, THUMPD1 Reader: — Eraser: —	High expression: NAT10	PCa cell line assay	35743017
**Ψ**	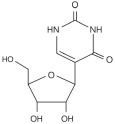	Writer: TRUB1/2, PUS1/7 Reader: — Eraser: —	—	—	—
**m^7^G**	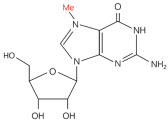	Writer: METTL1/WDR4, RNMT/RAM, WBSCR22/TRMT112 Relative: EIF3D/4A1, LARP1	High expression: METTL1	TCGA and CPTAC databases	34285249
**5mC**	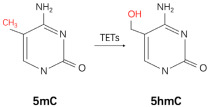	Writer: DNMT1/3A/3B Reader: MeCP2, MBD1/2/4, UHRF1/2 Cytosine Dioxygenase: TETs	Low expression: TET2 Low level: 5hmC	Clinical samples immunohistochemistry	26404510
**Histone modification**	—	Core histones: H2A, H2B, H3, H4	High expression: EZH2, LSD1, SET9	Clinical samples by immunohistochemistry	12374981 16079795 20959290
**Nucleosome positioning**	—	Chromatin remodeling complexes: SWI/SNF, ISWI, CHD, and INO80	CDH1 gene deletion	Clinical samples	22179824
**Higher-order chromatin organization**	—	Conformational molecules: CAF-1, CTCF, etc.	High expression: CAF-1 Low expression: CTCF	PCa cell line assay or clinical samples immunohistochemistry	19309489 31736271

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
