# Peer review of "Epigenetic Modifications in Prostate Cancer Metastasis and Microenvironment"

_cancers, 2023, doi:10.3390/cancers15082243_

Round 1

Reviewer 1 Report

The present Review article is focused on epigenetic modifications in prostate cancer metastasis and particularly on tumor remodeling and communication of tumor microenvironment, which is a very interesting field that could provide new treatments for this cancer type.

However, despite the outstanding and present-day topic, the manuscript is not well organized, it is quite confusing in many points and the aim is not clearly defined. These concerns make the manuscript difficult to read.

-In the Introduction section, the authors should also provide a brief description of the most involved (and recognized) molecular events occurring in PCa.

-The authors include two paragraphs describing alterations of DNA methylation and RNA modification, in general and in PCa. In this sectioning, the authors do not consider the other classes of epigenetic modifications (histone modifications and non-coding RNAs), which are poorly described within the later paragraphs (4-6). As an option, the authors could focus their Review just on RNA modifications (the title should be changed consequently).

-Table 1 (Epigenetic modification events in prostate cancer) does not summarize all the known epigenetic modifications occurring in PCa [see for example “López J et al Genes (Basel). 2022 Feb 18;13(2):378. doi: 10.3390/genes13020378”; but several recent Reviews have been published on this topic]. Nevertheless, Table 1 would be more interesting if it described the epigenetic modification with the related experimental evidence and included the proper literature reference(s).

-Figure is not enough informative and should be detailed, at least in the legend. 

Minor comments:

There are many language mistakes as well as many formatting errors. 

To cite some examples:

-page 2: “metathesize” should be replaced by “metastasize”

-paragraph 3.6 is erroneously in italic

-paragraph 2.7 at page 4 should be 3.7

Please recheck the whole text.

Author Response

Thanks for the valuable comment raised by this reviewer. To address these questions, we respond as followed:

  1. We have supplemented with the addition of PCa-related most involved (and recognized) molecular events in the introduction section.
  2. We expanded the content of chromatin-related regulation by adding 2.2. histone modification,2.3. nucleosome positioning,2.4. higher-order chromatin organization and added related content in sections 4 and 5.
  3. We updated Table 1 relevant evidence descriptions and references information.
  4. The legend information in Figure 1 is described in detail.

Finally, we recheck the whole text.

Reviewer 2 Report

Overall the review summarized DNA methylation and RNA modification as epigenetic modifications in prostate cancer metastasis and microenvironment. The better understanding of this aspect will guidance the clinical applications in prostate cancer. However to date, epigenetic regulation at least includes DNA/RNA modification, ncRNAs, and nucleosome positioning, higher-order chromatin organization etc. The authors only focused on the DNA methylation and RNA methylation, therefore, the scope should be expand to include more other aspects' contents to cover the scope of the title. Bellow are some comments to improve the manuscript.

1. Each section only includes a small paragraph and too brief and lack of comprehensive interpretation, should be more elaborate to enhance the the depth and impact for each section.

2. For elaboration of higher chromatin organization, for example, CTCF, the master regulator of 3D chromatin organization, is required for the higher-order chromatin maintenance, mutation (Li et al, Protein & Cell, 2017, 8(7)544-549) of methylation of the CTCF binding sites will affect CTCF binding thus lead to the alteration of the 3D genome architecture. alternative spliced isoform of CTCF can compete with the canonical CTCF binding thus affect the 3D architecture and gene expression causing cell apoptosis in cervical cancer derived cells (Li et al, Nature Communications, 2019, 10:1535). And CRISRi screens also reveals a DNA-methylation-mediated 3D-dependent causal mechanism in prostate cancer (Ahmed et al, Nat Communications, 2019, 12:1781). Which indicates the DNA methylation that affect the 3D-architecture will finally contribute to causal of prostate cancer.

3. Nucleosome positioning also a key epigenetic regulator of the genome, also plays multiple functions in multiple process, like somatic cell reprogramming (Huang et al., Sci Rep, 2015, 5:17691), differentiation and development etc. It should also contribute to prostate cancer, like the investigation of nucleosome patterns in circulating tumor DNA reveal transcriptional regulation of advanced prostate cancer phenotypes (Sarkar et al., Cancer Discovery, 2023, 13(3):632-653.).

4. P1, Paragraph 1 should include the reference when they cite the the progress from previous studies or reviews.

5. P3, in Section 3.4, "PRAD" should provide the full name.

6. Page 5, Section 5, "TME" should provide the full name.

7. Table 1 should insert a column with the corresponding references.

8. Increase the resolution of Table 1.

9. The format of Section 3.6 should be consistent with the rest of the manuscript.

Author Response

Thanks for the valuable comment raised by this reviewer. To address these questions, we respond as followed:

According to the comments, we have expanded the content of chromatin-related regulation by adding 2.2. histone modification,2.3. nucleosome positioning,2.4. higher-order chromatin organization and added related content in sections 4 and 5.

  1. We have expanded the overall content and provided appropriate overviews and summaries of the individual sections to enhance the depth and impact for each section.
  2. We have expanded the content of higher-order chromatin organization by adding 2.4. higher-order chromatin organization and added related content in sections 4.
  3. We have expanded the content of nucleosome positioning by adding 2.3. nucleosome positioning and added related content in sections 4.
  4. We updated the reference in P1, Paragraph 1.
  5. We provided the full name about “PRAD” in Section 3.4.
  6. We provided the full name about “TME” in Section 5.
  7. We added the references information in Table 1.
  8. We increased the resolution in Table 1.
  9. The format of Section 3.6 has been corrected.

Round 2

Reviewer 1 Report

The manuscript has been largely  improved

Author Response

Dear reviewer,

       Thank you very much for your recognition and have a great day!

Reviewer 2 Report

Though the revised version does not extensively cover the research progress in the epigenetic field, the contents have been substantially improved. A minor point is that: some of the conclusions are not only reported in one or two literatures. Please cite more related references when you summarized those points in respective of the similar findings. 

For the revised version in last sentence in page 1, "In addition to DNA methylation, chromosome-related regulatory means such as histone modification, nucleosome localization, and higher-order chromatin organization play an important role in the phenotypic transformation of prostate cancer." This conclusion comes from bunches of studies. Should included important findings and previously reviewed literatures.

Author Response

Thanks for the valuable comment raised by this reviewer. Based on your comments, we screened the full text and supplemented the reference list for the corresponding content. For example, “In addition to DNA methylation, chromosome-related regulatory means such as histone modification[10], nucleosome localization[11], and higher-order chromatin organization[12] play an important role in the phenotypic transformation of prostate cancer.”, “The dysregulation of RNA modification is based on the regulation of DNA methylation on the one hand[16], and is closely related to intercellular communication on the other hand[7].”, “Nucleosome remodeling is regulated by DNA methylation[35], histone modification[34], chromatin remodeling complexes[18], etc.”, and so on.